# The Curcumin Derivative GT863 Protects Cell Membranes in Cytotoxicity by Aβ Oligomers

**DOI:** 10.3390/ijms24043089

**Published:** 2023-02-04

**Authors:** Yutaro Momma, Mayumi Tsuji, Tatsunori Oguchi, Hideaki Ohashi, Tetsuhito Nohara, Naohito Ito, Ken Yamamoto, Miki Nagata, Atsushi Michael Kimura, Shiro Nakamura, Yuji Kiuchi, Kenjiro Ono

**Affiliations:** 1Division of Medical Pharmacology, Department of Pharmacology, School of Medicine, Showa University, Tokyo 142-8555, Japan; 2Division of Neurology, Department of Internal Medicine, School of Medicine, Showa University, Tokyo 142-8555, Japan; 3Pharmacological Research Center, Showa University, Tokyo 142-8555, Japan; 4Department of Hospital Pharmaceutics, School of Pharmacy, Showa University, Tokyo 142-8555, Japan; 5Department of Oral Physiology, School of Dentistry, Showa University, Tokyo 142-8555, Japan; 6Department of Neurology, Kanazawa University Graduate School of Medical Sciences, Kanazawa University, Kanazawa 920-8640, Japan

**Keywords:** Aβ oligomers, GT863, curcumin, Alzheimer’s disease, cell membranes, neurotoxicity

## Abstract

In Alzheimer’s disease (AD), accumulation of amyloid β-protein (Aβ) is one of the major mechanisms causing neuronal cell damage. Disruption of cell membranes by Aβ has been hypothesized to be the important event associated with neurotoxicity in AD. Curcumin has been shown to reduce Aβ-induced toxicity; however, due to its low bioavailability, clinical trials showed no remarkable effect on cognitive function. As a result, GT863, a derivative of curcumin with higher bioavailability, was synthesized. The purpose of this study is to clarify the mechanism of the protective action of GT863 against the neurotoxicity of highly toxic Aβ oligomers (Aβo), which include high-molecular-weight (HMW) Aβo, mainly composed of protofibrils in human neuroblastoma SH-SY5Y cells, focusing on the cell membrane. The effect of GT863 (1 μM) on Aβo-induced membrane damage was assessed by phospholipid peroxidation of the membrane, membrane fluidity, membrane phase state, membrane potential, membrane resistance, and changes in intracellular Ca^2+^ ([Ca^2+^]i). GT863 inhibited the Aβo-induced increase in plasma-membrane phospholipid peroxidation, decreased membrane fluidity and resistance, and decreased excessive [Ca^2+^]i influx, showing cytoprotective effects. The effects of GT863 on cell membranes may contribute in part to its neuroprotective effects against Aβo-induced toxicity. GT863 may be developed as a prophylactic agent for AD by targeting inhibition of membrane disruption caused by Aβo exposure.

## 1. Introduction

There is a rapidly aging population worldwide, and with aging, the prevalence of dementia worldwide continues to increase. Recent estimates predict that 50.4 to 65.1 million people worldwide were affected by dementia in 2019, with the prevalence increasing to 130.8 to 175.6 million by 2050 [1]. Approximately 60–80% of all dementias are Alzheimer’s disease (AD), and AD is the most common type of dementia known. Without effective treatments to control or prevent the progression of dementias, the incidence of AD will continue to increase, affecting health care, long-term care, and government (such as national medical expenses etc.). AD is characterized by neurofibrillary changes due to microtubule-associated protein tau and senile plaques due to extracellular deposition of amyloid β-protein (Aβ) [2]. Abnormalities in the production and accumulation of Aβ have been suggested to play a central role in the pathogenesis of AD [3]. In addition, soluble Aβ oligomers (Aβo), which are intermediates in the aggregation process, are more cytotoxic than Aβ monomers and are thought to cause AD [4]. Therapeutic strategies targeting Aβ or its aggregates have been pursued, and lecanemab, an anti-Aβo antibody, was rapidly approved by the United States Food and Drug Administration (US FDA) in 2023. Recently, high-molecular-weight (HMW) Aβo mainly composed of protofibrils, which is a toxic Aβo, have attracted attention [5,6]. Lecanemab binds Aβ aggregates, including oligomers and insoluble fibrils [7].

It has been reported that the consumption of foods high in polyphenol compounds may inhibit the development of AD [8]. Curcumin is a polyphenolic compound found in turmeric and other plants and has anticancer, anti-inflammatory, and antioxidant effects. Furthermore, curcumin has been shown to inhibit Aβ aggregation in vitro and crossed the blood–brain barrier, inhibited Aβ aggregation and fibrosis in the brain, and improved cognitive function in an experimental mouse model of AD [9]. However, in a study of 6-month treatment of patients with possible AD aged 50 years and older, no significant differences in cognitive function were reported between the treated and untreated groups [10]. The reason for the lack of significant results in this study was the poor bioavailability of curcumin. These results have led to the synthesis of curcumin analogues and derivatives that are expected to be very useful drugs for the treatment of AD [11]. GT863 (Figure 1) was synthesized as a derivative of curcumin; this derivative has higher bioavailability and brain translocation than curcumin [12]. In vivo, GT863 was reported to improve cognitive impairment in a mouse model of AD [13]. GT863 also showed dual inhibition of Aβ and tau aggregation and reduced the amount of aggregated Aβ and tau in the brains of senescence-accelerated mouse-prone 8 (SAMP8) [14]. Furthermore, GT863 was reported to be neuroprotective against hydrogen peroxide-induced cytotoxicity in PC12 cells [15].

One of the main challenges in the development of therapeutic agents for AD is the inhibition of oxidative stress and cell-membrane damage caused by highly cytotoxic Aβo exposure. In SH-SY5Y cells, Aβo exposure has been shown to increase reactive-oxygen-species (ROS) production and cause significant oxidative stress [6]. Oxidative stress has been implicated in the pathogenesis of various pathologies and human diseases—particularly neurodegenerative diseases, including AD [16,17]. Cell-membrane integrity is essential for cell function and survival. Chemical stimuli such as ROS induce peroxidation of plasma-membrane phospholipids, resulting in loss of plasma-membrane integrity [18]. Furthermore, the formation of pores of various sizes and properties in the plasma membrane also result in a loss of plasma-membrane integrity. Aβo exposure causes chemical disruption of the plasma membrane, forms ion-channel pores in the plasma membrane [19], and act similar to surfactants, disrupting the plasma-membrane lipid bilayer and injuring neurons [20]. However, there are no reports on the action of GT863 on cytotoxic Aβo-induced oxidative stress and cell-membrane damage.

In this study, we prepared the highly toxic HMW Aβo including mainly protofibrils, and the protective mechanism of GT863 against Aβo-induced neuronal injury was investigated, with a focus on the plasma membrane.

## 2. Results

### 2.1. Effects of Curcumin and GT863 on the Aggregation of Aβ _1–40_ and Aβ _1–42_

A thioflavin T (ThT) binding assay was used to compare the effects of curcumin and GT863 on the aggregation kinetics of Aβ_1-40_ and Aβ_1-42_. Figure 2A shows the amyloid-aggregation kinetics of Aβ_1-40_ (25 µM). In the absence of curcumin or GT863, the fluorescence intensity of Aβ_1-40_ increased exponentially without delay, and the maximum fluorescence intensity after 345 min reached approximately 1.8 times that at onset. However, in the presence of curcumin or GT863, the ThT fluorescence decreased in a concentration-dependent manner compared with Aβ_1-40_. The inhibition percent of Aβ_1-40_ aggregation reached approximately 80% after 345 min in the presence of 10 µM curcumin or GT863 (*n* = 6, Tukey’s test, curcumin; *p* < 0.0001, GT863; *p* < 0.0001).

Furthermore, Aβ_1-42_ aggregation kinetics were monitored for 120 min (Figure 2B). The aggregation of Aβ_1-42_ has been reported to cause severe neurotoxicity among all existing isoforms of Aβ [21]. Aβ_1-42_ alone aggregated faster than Aβ_1-40_, with an exponential increase in ThT fluorescence intensity without a delay phase and followed by a stationary phase. The saturation-fluorescence intensity of Aβ_1-42_ after 120 min reached approximately 3.5 times the fluorescence intensity observed at onset. Similar to the aggregation kinetics of Aβ_1-40_, the fluorescence of Aβ_1-42_ in the presence of curcumin or GT863 was reduced in a concentration-dependent manner. The inhibition percent of Aβ_1-42_ aggregation reached approximately 62% after 120 min in the presence of 10 µM curcumin or GT863 (*n* = 6, Tukey’s test, curcumin; *p* < 0.0001, GT863; *p* < 0.0001).

### 2.2. Effects of Curcumin and GT863 on Viability and Neurotoxicity in Aβo-Exposed Cells

#### 2.2.1. Changes in Viability Assessed with the MTT (3-(4,5-Dimethylthiazol-2-yl)-2,5-diphenyltetrazolium bromide) Assay

Preliminary experiments exposing SH-SY5Y cells to Aβo (1, 5, or 10 µM) for 3 h were performed using the MTT assay (Figure 3A). Cells exposed to experimental concentrations of Aβo for 3 h showed a significant reduction in cell viability compared with the control. The IC_50_ value for cell viability of Aβo-exposed SH-SY5Y cells for 3 h was 3.26 µM. Since there was no difference in cell viability between 5 µM and 10 µM Aβo at 3 h of incubation, we decided that 5 µM Aβo was a suitable concentration to induce cytotoxicity in SH-SY5Y cells (absorbance value of 5 µM Aβo: 0.660 ± 0.010, 10 µM Aβo: 0.622 ± 0.010, *n* = 8, Tukey, *p* = 0.4339). Therefore, 5 µM Aβo was used in the following experiments.

Curcumin and GT863 were lysed with 0.1% dimethyl sulfoxide (DMSO); therefore, 0.1% DMSO-treated cells were used as controls. There was no significant difference in viability between cells treated with 0.1% DMSO and untreated cells (absorbance value of 0.1% DMSO: 1.489 ± 0.02, untreated cells: 1.52 ± 0.02, *n* = 8, Tukey, *p* = 0.5087).

As shown in Figure 3B, treatment with curcumin and GT863 alone showed a significant increase in survival with 10 µM GT863 treatment compared with the control. The viability of Aβo-exposed cells was significantly decreased compared with that of the control cells, whereas Aβo-induced cytotoxicity was significantly suppressed by treatment with curcumin (1, 10 µM) and GT863 (1, 10 µM). No significant differences between curcumin and GT863 were found. As 10 µM GT863 treatment alone showed an effect on SH-SY5Y, 1 µM GT863 was considered appropriate for use in subsequent experiments.

#### 2.2.2. Staining with Calcein-AM/Ethidium Homodimer-1 (EthD-1)

Figure 4 shows SH-SY5Y cells exposed to Aβo in the presence of curcumin and GT863 stained with calcein-AM/EthD-1. The cytotoxicity of cells exposed to 1% saponin is expressed as 100%. As shown in Figure 4A, SH-SY5Y cells exposed to Aβo for 3 h showed significantly increased cytotoxicity compared with the control, and improvements in cytotoxicity were observed when curcumin or GT863 was used to treat the cells. However, there was no significant difference between the cytotoxicity of curcumin-treated and GT863-treated cells (*n* = 10, Tukey, *p* = 0.8478).

In the fluorescence images, the control cells showed more green fluorescence, and few cells with red-stained nuclei were observed (Figure 4F). In contrast, SH-SY5Y cells exposed to Aβo showed an increase in red-stained nuclei, indicating cytotoxic effects due to cell-membrane damage (Figure 4G). Curcumin and GT863 treatment reduced the number of red-stained nuclei compared with Aβo exposure (Figure 4H,I). Both curcumin and GT863 showed protective effects against Aβo-mediated cell injury.

### 2.3. ROS Production

Figure 5 shows ROS production in SH-SY5Y cells exposed to Aβo in the presence of curcumin and GT863.

As shown in Figure 5A, ROS production was significantly increased in SH-SY5Y cells exposed to Aβo for 30 min compared with the control. Curcumin and GT863 treatment significantly reduced ROS formation that was increased by Aβo exposure. The fluorescence enhanced by Aβo exposure was weakened by curcumin and GT863 treatment, indicating antioxidant activity of both curcumin and GT863. There was no significant difference between the ROS production of curcumin-treated and GT863-treated cells (*n* = 10, Tukey, *p* = 0.9993).

In the fluorescence images shown in Figure 5B,I, green dichlorofluorescein (DCF) fluorescence was increased in SH-SY5Y cells exposed to Aβo for 30 min compared with the control.

### 2.4. Effects of Curcumin and GT863 on Aβ-Induced Disruption of Membrane Integrity

Aβ is assumed to bind directly to the cell membrane, injuring the lipid-bilayer structure and entering the cell [22,23]. This study investigated changes in cell-membrane fluidity and phospholipid peroxidation of the cell membrane and cell-membrane potential following exposure to Aβo, curcumin, and GT863.

#### 2.4.1. Phospholipid Peroxidation in Cell Membranes

Increased phospholipid peroxidation in cell membranes contributes to cell-membrane damage [24].

Exposure of SH-SY5Y cells to Aβo for 30 min resulted in a significant increase in phospholipid peroxidation of the cell membrane. The increase in phospholipid peroxidation caused by Aβo exposure was significantly inhibited by curcumin and GT863, indicating an antioxidant effect of both curcumin and GT863 on the cell membrane (Figure 6A). There was no significant difference between the phospholipid peroxidation of the cell membrane of curcumin-treated and GT863-treated cells (*n* = 10, Tukey, *p* = 0.9934).

#### 2.4.2. Cell Membrane Fluidity

In SH-SY5Y cells exposed to Aβo for 30 min, the fluidity of the cell membrane was significantly reduced. However, treatment with curcumin and GT863 prevented the loss of membrane fluidity (Figure 6B). There was no significant difference between the cell-membrane fluidity of curcumin-treated and GT863-treated cells (*n* = 10, Tukey, *p* = 0.2574).

#### 2.4.3. Cell-Membrane Lipid Order

The phase state of the lipid bilayer (lipid order) is important for understanding the fluidity and stiffness of biological membranes. The liquid-ordered (Lo) phase shows green emission, whereas the liquid-disordered (Ld) phase shows red emission due to a long-wavelength shift. Images of green and red fluorescence were analyzed ratiometrically (red/green fluorescence ratio).

The cell membrane of SH-SY5Y cells exposed to Aβo showed a decreased Ld phase, as shown in Figure 6H, and a lower ratiometric fluorescent value (red/green), as shown in Figure 6L, than the control. Exposure to Aβo may have increased the lipid-membrane density of the cell membrane, resulting in a stiffer cell membrane. This suggests that the density of fat in the cell membrane increased, resulting in reduced fluidity. Curcumin (Figure 6E,I,M) and GT863 (Figure 6F,J,N) treatment suppressed the increased Lo-phase percentage due to Aβo exposure. Treatment with curcumin and GT863 restored membrane fluidity, and the phase state showed an increased proportion of Ld phase, stabilizing the cell membrane.

#### 2.4.4. Membrane Potential

Immediately after Aβo exposure, the membrane potential increased significantly compared with controls and remained high until 280 s (Tukey’s test, *p* < 0.0001). Treatment with curcumin and GT863 significantly suppressed the increase from 10 s after Aβo exposure (Tukey’s test, curcumin; *p* < 0.0001, GT863; *p* < 0.0001) (Figure 7A). Curcumin and GT863 had an electrically stabilizing effect on cell membranes.

#### 2.4.5. Whole-Cell Patch-Clamp Studies

Figure 7B,C shows changes in resting-membrane potential and membrane resistance by exposure to Aβo, curcumin, and GT863, as measured by the whole-cell patch-clamp technique.

Aβo exposure significantly increased the resting-membrane potential of SH-SY5Y cells. However, curcumin and GT863 treatment showed no significant difference in resting-membrane potential compared with Aβo exposure (Figure 7B).

Aβo exposure significantly reduced the electrical membrane resistance of SH-SY5Y cells, and curcumin and GT863 treatment improved the cell-membrane resistance that was reduced by Aβo exposure (Figure 7C). There was no significant difference between the cell-membrane resistance of curcumin-treated and GT863-treated cells (*n* = 10, Tukey, *p* = 0.9983).

### 2.5. Changes in Intracellular Calcium-Ion Concentrations ([Ca^2+^]i) following GT863 Treatment

Aβo exposure significantly increased [Ca^2+^]i in SH-SY5Y cells immediately after exposure. Aβo exposure caused [Ca^2+^]i to peak at 60–70 s, and sustained high [Ca^2+^]i values were maintained until 300 s.

To investigate whether this Ca^2+^ influx is mediated through calcium channels, we measured Aβo-stimulated changes in [Ca^2+^]i in the presence of L-type/T-type voltage-gated calcium-channel (VGCC) blockers (nicardipine), N-type VGCC blockers (PD173212), and T-type highly selective VGCC blockers (NNC55-0396). Pretreatment with nicardipine (10 µM) significantly suppressed the sustained elevation of [Ca^2+^]i from 60 to 300 s of Aβo exposure (Tukey’s test, *p* < 0.05) (Figure 8A). However, pretreatment with PD173212 (1 µM) and NNC55-0396 (8 µM) only slightly suppressed the sustained elevation of [Ca^2+^]i induced by Aβo exposure (after 300 s of Aβo exposure, PD173212; *p* = 1.0000, NNC55-0396; *p* = 0.5271) (Figure 8A). Under Ca^2+^-free conditions, Aβo slightly increased [Ca^2+^]i compared with controls only immediately after exposure. These results suggest that in SH-SY5Y cells, Aβo exposure promotes Ca^2+^ influx via L-type VGCCs.

The increase in [Ca^2+^]i in response to Aβo stimulation was inhibited by 1 µM curcumin and 1 µM GT863 pretreatment (Figure 8B). There was no significant difference between the [Ca^2+^]i levels of curcumin-treated and GT863-treated cells (after 300 s of Aβo exposure, ANOVA, *p* = 0.7906).

## 3. Discussion

In this study, the mechanism of action of GT863, a derivative of curcumin, on Aβo-induced neuronal-membrane damage was investigated. HMW Aβo are widely suggested to be the most toxic of the Aβ species and have been proposed as one of the etiologies of AD [4]. Soluble Aβ monomers undergo conformational changes and self-associate to form β-sheet-rich oligomers, which initiate polymerization and Aβ elongation. Inhibition of oligomerization and slowing of conversion may therefore be an effective therapeutic approach. A number of recent studies suggest that compounds that modulate Aβ aggregation and reduce these toxic oligomers show promise for drug development against AD. For example, plant polyphenols such as myricetin, quercetin, and gallic acid have previously been reported to have inhibitory effects on Aβ aggregation, including oligomerization [25]. In particular, curcumin was considered a promising therapeutic agent, but its low bioavailability was a problem. Therefore, there are efforts to improve its low bioavailability by developing derivatives and formulating them in combination with other compounds. GT863 has higher bioavailability than curcumin and has been reported to inhibit Aβ and tau aggregation in vitro [13,14]. In the present study, comparable to curcumin, GT863 showed dose-dependent inhibition of Aβ_1-40_ and Aβ_1-42_ aggregation at low concentrations (Figure 2). We used 25 µM Aβ in the ThT assay, and GT863 showed an inhibitory effect on Aβ aggregation at concentrations as low as 1 µM, indicating that GT863 inhibited aggregation even at a concentration 1/25th that of Aβ. In a previous study, GT863 also inhibited full-length tau aggregation in a concentration-dependent manner (0.1 µM to 10 µM) [13]. It is speculated that GT863 enhances the aggregation-inhibitory activity of tau by replacing the monocyclic aromatic-ring structure of curcumin with a bicyclic structure [12]. However, the in vivo inhibition of Aβ and tau aggregation in the brain depends on the rate at which curcumin and GT863 migrate into the brain. As curcumin has a less favorable rate of brain migration and GT863 has a many-fold higher rate of brain migration than curcumin [12], it is speculated that GT863 may also have a greater difference in its aggregation inhibitory effect on these peptides in the brain than curcumin.

In the cell-viability and cytotoxicity experiments in our study, treatment with low concentrations of GT863 increased cell viability and decreased cytotoxicity compared with Aβo exposure alone (Figure 3B and Figure 4). Aβo exposure induced a disruption of membrane integrity and increased [Ca^2+^]i levels (Figure 6 and Figure 8). Extracellular Aβ encounters the lipid-bilayer barrier of the plasma membrane. The mechanism of Aβ-mediated membrane disruption depends on the Aβ structure, and very different effects on the membrane have been identified when comparing Aβ monomers, oligomers, or fibrils [20]. Aβ monomers have little effect on the membrane structure, but only Aβ oligomers form ion-channel pores in the plasma membrane [19]. It has been proposed that positively charged amino-acid residues of Aβ monomers bind to negatively charged cell-membrane phospholipids; after this, the Aβ conformation changes, the proportion of β-sheet structures increases, and the amyloid aggregates form [26]. On the other hand, Aβo have been reported to have the surfactant-like properties of amphiphilic peptides, which remove lipids from the cell membrane, thinning the cell membrane and disrupting it [27]. Concerning the assembly of extracellular Aβo, GT863 inhibited Aβ aggregation at low concentrations. Therefore, GT863 can be expected to reduce Aβo contact with the cell membrane.

Furthermore, oxidative stress is a causative factor for damage to cell membranes. In the present experiments, Aβo exposure resulted in increased peroxidation of plasma-membrane phospholipids and ROS production (Figure 5 and Figure 6A). Oxidative stress is a state of imbalance between the production of ROS and the antioxidant capacity of cells due to increased generation of ROS or dysfunction of the antioxidant system [28]. Biochemical changes induced by oxidative stress have been implicated in the pathogenesis of various human diseases—particularly neurodegenerative diseases, including AD [16,17]. Therefore, reducing oxidative stress and promoting neuronal survival with antioxidants may contribute to the prevention and early treatment of AD. In this experiment, it was confirmed that GT863 prevented Aβo-induced oxidative stress. The antioxidant properties of curcumin are derived from its β-diketone structure and phenolic groups, which curcumin uses to scavenge free radicals [29]. A randomized clinical trial involving patients treated with curcumin supplements showed that curcumin may reduce malondialdehyde levels and increase total antioxidant capacity [30]. However, in GT863, the β-diketone structure of curcumin is replaced by a pyrazole group, and the two phenol groups are replaced by an indol ring and a pyridylmethoxy group. No experiments were conducted in this study with regard to free-radical removal; thus, further research is needed.

Furthermore, GT863 inhibited Aβo-induced cell-membrane damage such as cell-membrane lipid peroxidation and reduction of membrane fluidity (Figure 6A). In AD, extracellularly and intracellularly formed Aβ aggregates can induce mechanical-membrane damage and promote lipid peroxidation. In neuronal-cell membranes, phospholipid portions undergo chain reactions with free radicals to produce various lipid peroxides [16,31]. These lipid peroxides bind to several membrane proteins, altering their structure and function and resulting in neurotoxicity [24]. In the presence of ROS, oxidation of polyunsaturated fatty acids (i.e., lipid peroxidation) releases damaged lipid fragments, ultimately resulting in the loss of protoplasmic-membrane integrity. Oxidized membranes promote protein aggregation at the cell surface, further exacerbating plasma-membrane damage [32]. However, membrane damage is prevented by antioxidants and the central lipid-repair factor glutathione peroxidase 4 (GPX4). In the present experiment, Aβo-induced peroxidation of phospholipids in the plasma membrane was inhibited by GT863, and it also had an inhibitory effect on ROS (Figure 6A).

The cell membrane is altered by Aβs to alter membrane fluidity, which plays an important role in the pathogenesis of AD. Aβo exposure causes not only reduced membrane fluidity and increased elasticity but also increased permeability, leading to regulation or elevation of intracellular ion concentrations and ultimately cell death [33]. The reduction of membrane fluidity caused by Aβo exposure is therefore considered to be closely related to cell injury. Indeed, hippocampal membranes of patients with AD showed significantly lower fluidity than those of older people without dementia [34]. It has also been shown that Aβ_1-42_ oligomers can alter the bending rigidity of the lipid bilayer [33]. Considering that polyunsaturated fatty acids are abundant in the central nervous system [35,36], cell-membrane fluidity and membrane-phase state are strongly associated with neuronal injury, including in AD. In this experiment, SH-SY5Y cells exposed to Aβo showed decreased cell-membrane fluidity and a higher proportion of the Lo phase, which has a denser lipid membrane, than control cells. However, treatment with GT863 restored membrane fluidity, increased the proportion of the Ld phase in the lipid membrane, and stabilized the cell membrane (Figure 6B,C). Considering that GT863 also strongly inhibited Aβ aggregation (Figure 2), it is possible that GT863 binds to Aβo, inhibiting the direct effect of Aβo on the membrane and improving the loss of fluidity.

Furthermore, Aβ binding to intracellular cholesterol has been reported to form Aβ-oligomeric Ca^2+^ channels at the cell membrane in a cholesterol-dependent manner [37]. In the present experiment, the addition of Aβo resulted in a sustained and remarkable increase in [Ca^2+^]i (Figure 8). Intracellular Ca^2+^ homeostasis plays a pivotal regulatory role in many aspects of neurophysiology. Abnormal intracellular Ca^2+^ contributes to pathophysiology, such as necrosis and apoptosis [38], and is considered to be an important pathophysiological factor in AD [39]. Sustained elevated [Ca^2+^]i causes excessive Ca^2+^ loading of mitochondria, increasing the production of ROS and ultimately prompting cell death [40]. Increased Ca^2+^ influx into cells due to the addition of Aβo was significantly inhibited by nicardipine pretreatment, but no significant difference was shown by NNC 55-0396 and PD173212 pretreatment (Figure 8A). Therefore, the increase in Ca^2+^ entry into cells upon Aβo addition appears to be mediated by L-type VGCCs. Aβo exposure may enhance Ca^2+^ influx via L-type VGCCs; furthermore, Aβo may increase [Ca^2+^]i by forming Ca^2+^-permeable channels and pores in the plasma membrane [37]. GT863 inhibited the sustained [Ca^2+^]i increase induced by Aβo exposure (Figure 8B). As shown in Figure 2, GT863 inhibited Aβ aggregation, suggesting that GT863 bound to Aβo and inhibited the effects of Aβo on VGCCs. Therefore, we suggest that GT863 reduced cytotoxicity by suppressing the increase in intracellular Ca^2+^.

Experiments using the membrane-potential-sensitive dye DiBAC_4_(3) showed changes in membrane potential due to Aβo exposure (Figure 7A). The application of Aβo to SH-SY5Y cells induced depolarization, which was detected by DiBAC_4_(3). DiBAC_4_(3) was taken up intracellularly by depolarization, and a rapid increase in membrane potential was observed immediately after Aβo exposure compared with the control, followed by a sustained increase (Figure 7A). Curcumin and GT863 inhibit excessive depolarization of the plasma membrane by Aβo, stabilize the plasma membrane, maintain [Ca^2+^]i homeostasis, and may be protective against neuronal-membrane injury.

In this in vitro experiment, curcumin and GT863 showed almost the same degree of protection against Aβo-induced cell injury (Figure 3). Curcumin has very good pharmacological effects as an anticancer, anti-inflammatory, and antioxidant agent as well as anti-AD effects, such as inhibition of Aβ aggregation and tau aggregation [11,41]. Curcumin has been shown to be protective not only for neurons but also for astrocytes, which play an important role in brain homeostasis [42]. Curcumin is lipophilic and therefore passes through the blood–brain barrier and binds to amyloid [9,43]; however, its brain-translocation rate is not favorable [44,45]. Furthermore, much of the orally ingested curcumin is metabolized in the small intestine and liver, resulting in trace amounts of the free type of curcumin ultimately entering the bloodstream [45,46,47], which has the weakness of low bioavailability. GT863 was developed as a derivative of curcumin, overcoming its weakness by improving the bioavailability while retaining its inhibitory effect against Aβ aggregation. The pharmacokinetics of GT863 were determined following oral administration of 50 mg/kg to normal Sprague Dawley rats, with blood concentrations at 3 h approximately nine times higher than those of curcumin and brain concentrations 20 times higher than those of curcumin [12]. Furthermore, in a study in which GT863 was administered orally to normal mice at a dose of 40 mg/kg, the concentration of GT863 in the blood and brain was measured over time. The maximum blood concentration of GT863 was 2.005 ± 0.267 µg/mL at 3 h (mean ± SEM, *n* = 4), and the maximum brain concentration at 6 h was 1.428 ± 0.413 µg/g (mean ± SEM, *n* = 4) at 6 h; the respective area-under-the-curve (AUC) values were 16.24 µg-h/mL and 13.03 µg-h/mL, with a brain-to-plasma ratio of 0.80 [13]. From these results, it appears that GT863 crosses the blood–brain barrier and acts directly on neurons. Namely, although no difference in the protective effects of curcumin and GT863 was observed in in vitro experiments, it can be inferred that GT863 is more protective against Aβ-induced neuronal injury at the same dose of administration because it is more likely to be transferred to the central-nervous-system tissue due to its improved bioavailability. Therefore, GT863 can be expected to prevent the onset and progression of AD in the future. However, given the limitations of this study, further studies using different cell types, such as non-neuronal cell lines, are needed to confirm the efficacy of GT863.

## 4. Materials and Methods

### 4.1. Drugs and Reagents

Human Aβ_1-42_ was purchased from Peptide Institute Inc. (Osaka, Japan), curcumin (Figure 1) from Thermo Fisher Scientific K.K. (Waltham, MA, USA), and GT863 (purity: 99.1%) (Figure 1) from Green Tech Co., Ltd. (Kyoto, Japan). Nicardipine was obtained from Merck KGaA, NNC 55-0396 from Cayman Chemical Company (Ann Arbor, MI, USA), and PD173212 from Abcam PLC. (Cambridge, UK). Curcumin and GT863 were dissolved in DMSO, with a final concentration of 0.1%. The other chemical reagents used were commercially available special-grade products.

### 4.2. Aggregation Kinetics of Aβ_1-40_ and Aβ_1-42_

Aβ_1-40_ and Aβ_1-42_ aggregation kinetics were measured using the SensoLyte Thioflavin T β-Amyloid (_1-40_, _1-42_) Aggregation Kit (AS-72213, AS-72214; AnaSpec, Inc., Fremont, CA, USA). The higher the thioflavin-T (ThT) fluorescence intensity of the aggregation kinetics of Aβ_1-40_ or Aβ_1-42_, the more Aβ-fibril formation. Curcumin and GT863 were added, respectively, and shaken at 37 °C before measurement, and the fluorescence signal of ThT was measured every 15 min using a Spectra Max i3 (Molecular Devices, Sunnyvale, CA, USA) at excitation and emission wavelengths of 440 nm and 484 nm. The final concentration of Aβ (_1-40_, _1-42_) was 25 µM, and of curcumin and GT863 were 1 µM, 5 µM, and 10 µM, respectively. The fluorescence intensity at onset of the measurement was expressed as 100%.

### 4.3. Preparation of Aβo

For preparation of Aβo as the most toxic amyloid species, Aβ_1-42_ was dissolved in 10 mM NaOH and sonicated for 2 min, and then phosphate buffer (PBS) was added. The Aβ solution was filtered through a 0.22 µm-polyvinylidenedifluoride (PVDF) filter (Merck KGaA) and incubated at 37 °C for 1 h. After incubation, it was centrifuged at 16,000× *g* for 5 min and the supernatant was collected to determine the protein concentration using Bio-Rad Protein Assay Dye Reagent Concentrate (Bio-Rad Laboratories, Inc, Hercules, CA, USA). The obtained Aβo was diluted to 50 µM in PBS and stored at −80 °C. Some of the preparations were confirmed for the presence of HMW Aβo, including protofibrils using size-exclusion chromatography (SEC) and transmission electron microscopy (TEM). Samples were fractionated on a Superdex 75 increase 10/300 GL (Cytiva, Tokyo, Japan) at a flow rate of 0.8 mL/min using 10 mM phosphate buffer, pH 7.4. As shown in Appendix A, the peak of Aβo was at 10.02 min. We also confirmed the morphology of Aβo using an H-7600 transmission electron microscope (TEM; Hitachi, Ltd., Tokyo, Japan), and HMW Aβo containing many protofibrils was observed in the Aβo solution, as previously reported [48] (Appendix A).

### 4.4. Cell Culture and Drug Treatment

SH-SY5Y cells (human neuroblastoma, EC-94030304) were obtained from the European Collection of Authenticated Cell Cultures (London, UK). Cells were cultured in DMEM/Ham’s F-12 (FUJIFILM Wako Pure Chemical Corporation, Osaka, Japan) containing 10% fetal bovine serum (FBS) and antibiotics (penicillin G sodium salt, streptomycin sulfate, amphotericin B) and maintained in humidified conditions of 5% CO_2_ and 95% air at 37 °C. SH-SY5Y cells resemble neurons in terms of morphological and neurochemical properties. These cells have been widely used to assess neuronal injury or cell death, including in degenerative diseases. SH-SY5Y cells were treated with All-Trans-Retinoic Acid (FUJIFILM Wako Pure Chemical Corporation) at a final concentration of 10 µM and cultured in medium containing 10% FBS for 7 days to induce differentiation. Considering that Aβo may promote polymerization and degradation by incubation at 37 °C, Aβo exposure to SH-SY5Y cells was limited to 3 h.

### 4.5. Assay of Viability and Cytotoxicity in SH-SY5Y Cells

#### 4.5.1. Cell-Viability Assay

Assessment of the effects of curcumin and GT863 on the cell viability of SH-SY5Y cells exposed to Aβo was measured using the MTT cell-counting kit (Nacalai Tesque, Inc., Kyoto, Japan). First, a preliminary experiment was performed in which differentiated SH-SY5Y cells (1 × 10^6^ cells/mL) were cultured in collagen-coated 96-well plates and exposed to Aβo (1, 5, 10 µM) for 3 h. As shown in Figure 3A, the appropriate Aβo concentration to induce cytotoxicity was 5 µM.

Next, to investigate the protective effects of curcumin and GT863 on Aβo-induced cytotoxicity, SH-SY5Y cells were treated with Aβo + curcumin (1, 10 µM) or Aβo + GT863 (1, 10 µM) for 3 h. After incubation, the MTT assay was performed and measured at 570 nm using a Spectra Max i3 (Molecular Devices) microplate reader.

#### 4.5.2. Calcein-AM and EthD-1 (Live/Dead) Cell Assay

Live cells and dead cells were also observed by calcein-AM and EthD-1 co-staining. SH-SY5Y cells were cultured at 1 × 10^6^ cells/mL in 96-well plates and treated with Aβo (5 µM), Aβo + curcumin (1 µM), or Aβo + GT863 (1 µM) for 3 h. After incubation, the cells were co-stained with calcein-AM and EthD-1(Thermo Fisher Scientific K.K.). Non-fluorescent calcein-AM is hydrolyzed by intracellular esterases and emits green fluorescence depending on the number of live cells. EthD-1 only enters cells with damaged membranes and binds to nucleic acids, emitting bright-red fluorescence according to the number of dead cells. Cell fluorescence was measured using a Spectra Max i3 (Molecular Devices) at an excitation wavelength of 495 nm and emission wavelengths of 530 nm and 645 nm. In addition, the morphology of individual cells was assessed by observation under a fluorescence microscope (BZ-X800; Keyence, Osaka, Japan).

### 4.6. Detection of Reactive Oxygen Species (ROS)

5-(and-6)-chloromethyl-2’,7’ -dichlorodihydrofluorescein diacetate (CM-H_2_DCFDA: Thermo Fisher Scientific) was used to measure ROS. CM-H_2_DCFDA is a cell-permeable non-fluorescent substance but emits green fluorescence in the presence of ROS. SH-SY5Y cells were seeded onto 96-well plates at a concentration of 1 × 10^6^ cells/mL and treated with Aβo, Aβo + curcumin, or Aβo + GT863 for 30 min. Fluorescence intensity was measured at excitation and emission wavelengths of 488 nm and 525 nm, respectively, using a Spectra Max i3 (Molecular Devices). The cells were also observed under a fluorescence microscope (BZ-X800; Keyence) for visual oxidative-stress assessment.

### 4.7. Reaction to Cell Membrane

#### 4.7.1. Detection of Phospholipid Peroxidation in Cell Membranes by Diphenyl-1-pyrenylphosphine (DPPP)

To assess phospholipid peroxidation in the cell membrane, SH-SY5Y cells were stained with DPPP (Thermo Fisher Scientific). DPPP is a fluorogenic reagent with high reaction selectivity for hydroperoxides. SH-SY5Y cells (1 × 10^6^ cells/mL) were seeded onto 96-well plates and stained with 5 µM DPPP reagent at 37 °C for 10 min. The stained SH-SY5Y cells were treated with Aβo, Aβo + curcumin, or Aβo + GT863 for 30 min and the fluorescence intensity of DPPP oxide was measured using a Spectra Max i3 (Molecular Devices) at excitation and emission wavelengths of 352 nm and 380 nm.

#### 4.7.2. Membrane Fluidity

The membrane fluidity of SH-SY5Y cells was measured using the lipophilic pyrene probe pyrene decanoate (PDA) in the Membrane Fluidity Kit (ab189819, Marker Gene Technologies, Inc., Eugene, OR, USA). SH-SY5Y cells were seeded onto collagen-coated 96-well black plates at 1 × 10^6^ cells/mL; treated with Aβo, Aβo + curcumin, or Aβo + GT863 for 3 h; and then stained with a fluorescent lipid reagent containing PDA. When PDA forms excimers from monomers in spatial motion, the emission spectrum of PDA shifts to red. The ratio of fluorescence emission at 400 and 470 nm from 360 nm excitation (monomer-to-excimer ratio) was measured using a Spectra Max i3 (Molecular Devices) and was used as an estimate of membrane fluidity.

#### 4.7.3. Membrane-Lipid Order-Imaging Dye

The lipid order of the membrane in SH-SY5Y cell was visually confirmed using LipiORDER (Funakoshi Co., Ltd., Tokyo, Japan). The LipiORDER reagent shows green fluorescence in the Lo phase, where the lipid density of the cell membrane is high, whereas it shows red fluorescence in the Ld phase, where the density is low. SH-SY5Y cells were cultured in poly-D-lysine-coated glass-bottom dishes (MatTek Corp., Ashland, MA, USA) and treated with Aβo, Aβo + curcumin, and Aβo + GT863 for 3 h. After washing with PBS, the cells were treated with LipiORDER reagent. Stained cell membranes and intracellular membranes were imaged with a fluorescence microscope (A1 Confocal Laser Microscope System; Nikon Co., Tokyo, Japan) at two wavelengths (excitation: 402.9 nm, green fluorescence: 500–550 nm, red fluorescence: 570–620 nm). Green and red fluorescence images were analyzed ratiometrically (red/green fluorescence ratio) to visualize the lipid order in pseudo color.

#### 4.7.4. Changes in Lipid-Membrane Potential Induced by DiBAC_4_ (3)

Changes in membrane potential were measured using the Bis (1,3-dibutylbarbituric acid) trimethine oxonol sodium-salt (DiBAC_4_ (3)) reagent (Dojindo Molecular Technologies, Inc., Kumamoto, Japan). DiBAC_4_ (3), a membrane-potential-sensitive dye, is a pigment taken up into the cytoplasm when the plasma membrane is depolarized. The incorporated dye binds to intracellular proteins and intracellular membranes, causing fluorescence-intensity enhancement. SH-SY5Y cells were seeded onto 96-well black plates at a concentration of 5 × 10^5^ cells/mL and incubated with assay buffer (pH7.4; 20 mM HEPES, 120 mM NaCl, 2 mM KCl, 2 mM CaCl_2_, 1 mM MgCl_2_, 5 mM D-glucose) containing DiBAC_4_ (3) for 30 min at 37 °C. The cells were then exposed to Aβo, Aβo + curcumin, or Aβo + GT863. Changes in membrane potential were measured every 10 s for 300 s at excitation and emission wavelengths of 490 nm and 516 nm, respectively, using Flex Station 3 (Molecular Devices). Fluorescence intensity immediately before administration was set at 100%.

#### 4.7.5. Whole-Cell Patch-Clamp Recording

SH-SY5Y cells plated on poly-D-lysine-coated glass-bottom dishes were grown to 50–70% confluent. SH-SY5Y cells were treated with Aβo, Aβo + curcumin, or Aβo + GT863 for 30 min. All membrane potentials were recorded in a current-clamp configuration using a Multiclamp 700B amplifier (Molecular Devices), following the method previously described [6]. Input resistance was determined from recordings of the voltage response to 300 ms hyperpolarizing, 10 pA current steps.

### 4.8. Intracellular Calcium-Concentration ([Ca^2+^]i) Measurement

[Ca^2+^]i was measured using the FLIPR Calcium 5 Assay Kit (Molecular Devices). SH-SY5Y cells were loaded with FLIPR reagent diluted in buffer (pH 7.4; 140 mM NaCl, 2.7 mM KCl, 1.8 mM CaCl_2_, 12 mM NaHCO_3_, 5.6 mM D-glucose, 0.49 mM MgCl_2_, 0.37 mM NaH_2_PO_4_, 25 mM HEPES) for 60 min at 37 °C. Aβo, Aβo + curcumin or Aβo + GT863 was then added to the loaded cells and the changes in [Ca^2+^]I were measured using a Flex Station 3 (Molecular Devices) with FLIPR fluorescence signals at excitation and emission wavelengths of 485 nm and 525 nm, respectively, every 5 s for 300 s. The fluorescence intensity immediately before administration was expressed as 100%. In addition, changes in [Ca^2+^]i with and without Aβo were also measured in the absence of extracellular ionized calcium (pH 7.4; 140 mM NaCl, 2.7 mM KCl, 12 mM NaHCO_3_, 5.6 mM D-glucose, 0.49 mM MgCl_2_, 0.37 mM NaH_2_PO_4_, 25 mM HEPES, 100 mM EGTA). To confirm Ca^2+^ influx through calcium channels by Aβo exposure, L-type/T-type calcium-channel antagonist (nicardipine: 10 µM), N-type calcium-channel antagonist (PD173212: 1 µM), or T-type highly selective calcium-channel antagonist (NNC 55-0396: 8 µM) was pretreated 5 min before Aβo addition and changes in [Ca^2+^]i were measured.

### 4.9. Statistical Analysis

Each measurement was repeated three times. Results of cellular experiments were expressed as mean ± S.E.M. values. The effects of the various treatments were compared with SH-SY5Y cells treated with 0.1% DMSO, which is also included in other reagents, using a one-way analysis of variance (ANOVA) followed by a Tukey or Dunnett post hoc test. For all tests, a value of *p* < 0.05 was considered statistically significant.

## 5. Conclusions

GT863 improved the cell viability of SH-SY5Y cells injured by Aβo, which include HMW Aβo, mainly composed of protofibrils. GT863 inhibited plasma-membrane phospholipid peroxidation, reduced excessive depolarization, and stabilized the plasma membrane against Aβo exposure-induced plasma-membrane injury. Furthermore, GT863 inhibited excessive Ca^2+^ entry into cells upon Aβo exposure and maintained [Ca^2+^]i homeostasis (Figure 9). The exact molecular mechanism of the membrane-protective effect of GT863 remains unclear, but this effect may partially contribute to the neuroprotective effect of GT863 with regard to Aβo-induced toxicity. The study suggested that GT863 could be developed as a prophylactic and progression-preventive agent for AD by targeting the inhibition of Aβo-induced membrane disruption.

## Figures and Tables

**Figure 1 ijms-24-03089-f001:**
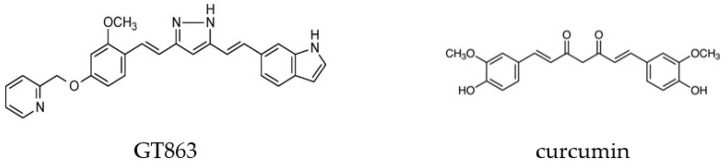
Structures of GT863 and curcumin.

**Figure 2 ijms-24-03089-f002:**
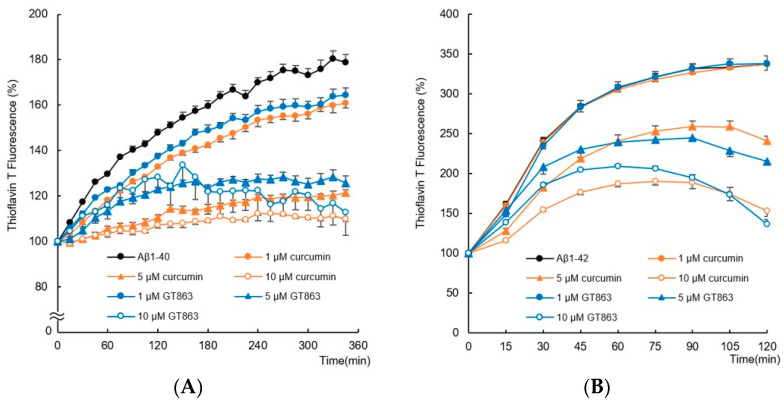
Inhibitory effect of GT863 and curcumin on Aβ_1-40_ and Aβ_1-42_ aggregation kinetics as monitored by the thioflavin-T fluorescence assay. The fluorescence intensity was evaluated with the value immediately before administration as 100%. (**A**) Aβ_1-40_ (25 µM) with 0.1% DMSO, curcumin (1, 5, 10 µM), or GT863 (1, 5, 10 µM). (**B**) Aβ_1-42_ (25 µM) with 0.1% DMSO, curcumin (1, 5, 10 µM), or GT863 (1, 5, 10 µM). The *p*-values in ANOVA were < 0.001. Each value expresses the mean ± S.E.M. (*n* = 6).

**Figure 3 ijms-24-03089-f003:**
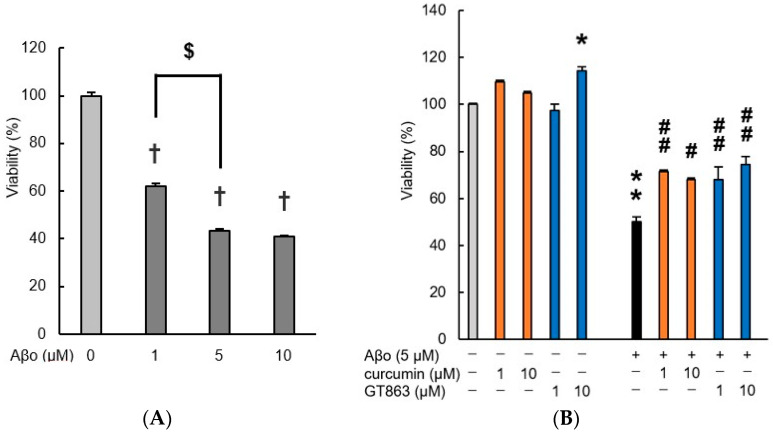
Cell viability of SH-SY5Y cells. Cell viability was evaluated by MTT assay. (**A**) Cell viability of SH-SY5Y cells exposed to Aβo (1,5, 10 µM) for 3 h. Each value is expressed relative to the viability of control (set to 100%). In the absence of Aβo, the absorbance value of control for 3 h was 1.521 ± 0.0184 (mean + S.E.M). †, *p* < 0.0001 for control versus Aβo-exposed cells (*n* = 8, Tukey). $, *p* < 0.0001 for 1 µM Aβo-exposed versus 5 µM Aβo-exposed cells (*n* = 8, Tukey). (**B**) Cell viability of SH-SY5Y cells exposed to 5 µM Aβo and treated with Aβo + curcumin (1, 10 µM), Aβo + GT863 (1, 10 µM) for 3 h. Each value is expressed relative to the viability of control cultures (set to 100%). +: inclusion of 5 µM Aβo, curcumin, GT863; −: non-inclusion. The *p*-values in ANOVA were < 0.001. Each value expresses the mean + S.E.M. of at least 3 independent experiments. In the absence of Aβo, the absorbance value of control for 3 h was 1.243 ± 0.068 (mean ± S.E.M). * *p* < 0.05, ** *p* < 0.0001 vs. control cells. # *p* < 0.05, ## *p* < 0.01 vs. Aβo-exposed cells (*n* = 8, Tukey).

**Figure 4 ijms-24-03089-f004:**
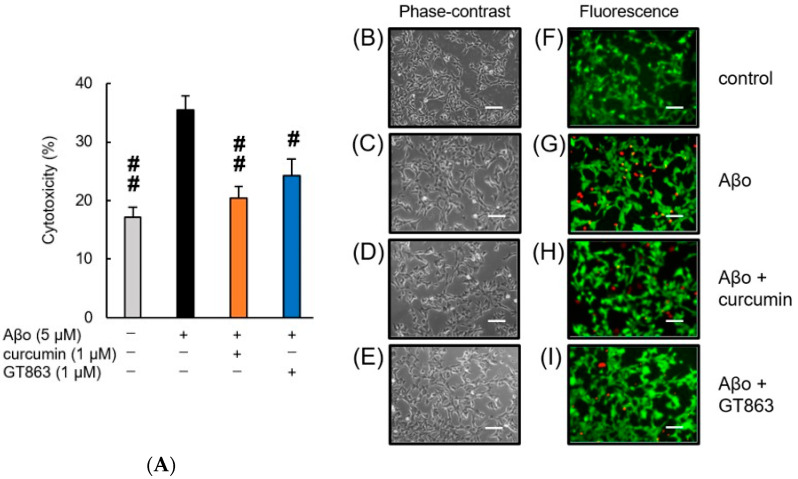
Effect of curcumin and GT863 on the cytotoxicity in Aβo-stimulated SH-SY5Y cells. The cytotoxicity in Aβo-stimulated SH-SY5Y cells was evaluated using EthD-1 cell assay. (**A**) The cytotoxicity of SH-SY5Y cells exposed to 5 µM Aβo and treated with Aβo + 1 µM curcumin, Aβo + 1 µM GT863. +: inclusion of 5 µM Aβo, curcumin, GT863; −: non-inclusion. The *p*-values in ANOVA were < 0.001. Each value expresses the mean + S.E.M. of at least 3 independent experiments. In the absence of 5 µM Aβo, the cytotoxicity of control, 1 µM curcumin-treated, 1 µM GT863-treated cells were 17.15 ± 1.67, 19.23 ± 1.74, and 19.11 ± 1.41%, respectively (no significant difference, *n* = 10, Tukey). #, *p* < 0.01, ##, *p* < 0.0001 for Aβo-exposed cells versus the other treated cells (*n* = 10, Tukey). (**B**–**I**) SH-SY5Y cells stained with calcein AM and Ethdium homodimer-1 observed using phase-contrast (**B**–**E**) and fluorescence microscopy (**F**–**I**). (**B**,**F**) Untreated SH-SY5Y cells; (**C**,**G**) SH-SY5Y cells exposed to 5 µM Aβo; (**D**,**H**) SH-SY5Y cells treated with 5 µM Aβo + 1 µM curcumin; (**E**,**I**) SH-SY5Y cells treated with 5 µM Aβo + 1 µM GT863. The scale bars represent 100 µm.

**Figure 5 ijms-24-03089-f005:**
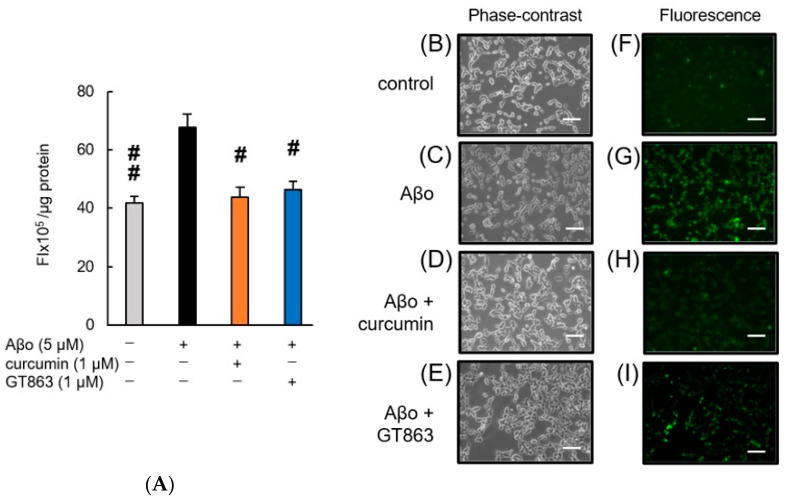
Effect of curcumin and GT863 on ROS generation in Aβo-stimulated SH-SY5Y cells. (**A**–**I**) ROS generation in Aβo-stimulated SH-SY5Y cells was evaluated using CM-H_2_DCFDA. (**A**) ROS generation in SH-SY5Y cells exposed to 5 μM Aβo and treated with Aβo + curcumin or Aβo + GT863. The *p*-values in ANOVA were 0.0002. Each value expresses the mean + S.E.M. of at least 3 independent experiments. In the absence of 5 µM Aβo, ROS levels of 1 µM curcumin-treated and 1 µM GT863-treated cells were 40.63 ± 4.78 and 44.90 ± 4.44 fluorescence intensity × 10^5^/µg protein, respectively (no significant difference, *n* = 10, Tukey). #, *p* < 0.05, ##, *p* < 0.01 for Aβo-exposed cells versus the other treated cells (*n* = 10, Tukey). (**B**–**E**) The SH-SY5Y cells observed using a phase-contrast microscope. (**E**–**H**) Fluorescence-microscopic images in SH-SY5Y cells were acquired using an inverted fluorescence microscope. (**B**,**F**) Untreated SH-SY5Y cells; (**C**,**G**) SH-SY5Y cells exposed to 5 µM Aβo; (**D**,**H**) SH-SY5Y cells treated with 5 µM Aβo + 1 µM curcumin; (**E**,**I**) SH-SY5Y cells treated with 5 µM Aβo + 1 µM GT863. The scale bars represent 100 μm.

**Figure 6 ijms-24-03089-f006:**
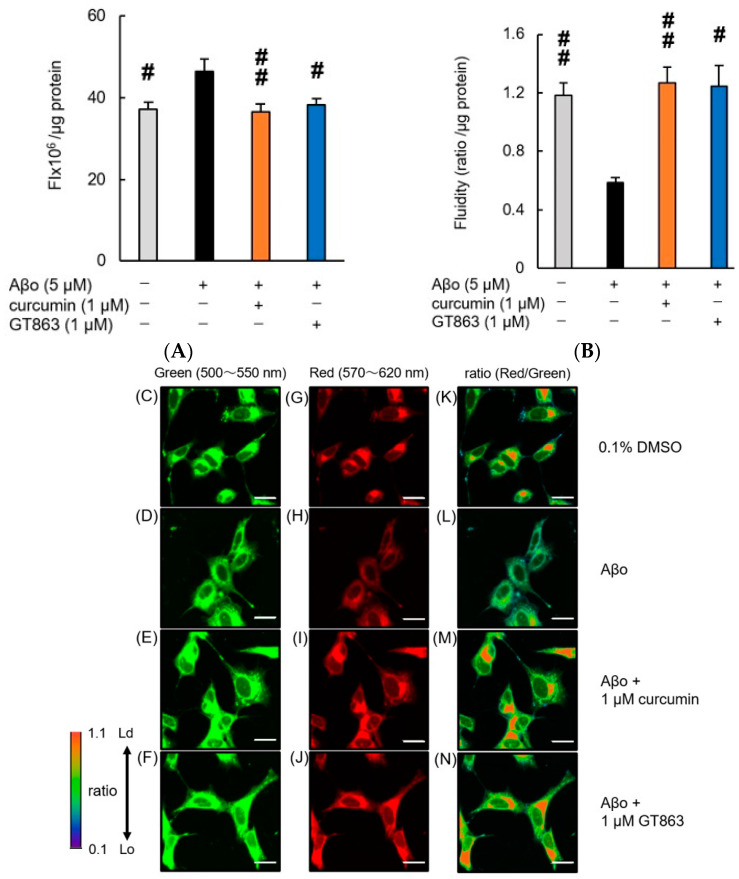
Effect of curcumin and GT863 on membrane-phospholipid peroxidation levels, cell-membrane fluidity, and membrane-lipid order-imaging dye. (**A**) The levels of membrane-phospholipid peroxidation in Aβo-stimulated SH-SY5Y cells was evaluated using DPPP. The levels of membrane-phospholipid peroxidation in SH-SY5Y cells exposed to 5 µM Aβo and treated with Aβo + 1 µM curcumin or Aβo + 1 µM GT863. In the absence of 5 µM Aβo, the membrane-phospholipid peroxidation levels of control, 1 µM curcumin-treated, and 1 µM GT863-treated cells were 33.32 ± 2.48, and 31.24 ± 1.72 fluorescence intensity × 10^6^/µg protein, respectively (no significant difference, *n* = 10, Tukey). +: inclusion of 5 µM Aβo, 1 µM curcumin, 1 µM GT863; −: non-inclusion. The *p*-values in ANOVA were < 0.001. Each value expresses the mean + S.E.M. of at least 3 independent experiments. # *p* < 0.05, ## *p* < 0.01 for Aβo-exposed cells versus the other treated cells (*n* = 10, Tukey). (**B**) The fluidity of the cell membrane in Aβo-stimulated SH-SY5Y cells was evaluated using PDA. The fluidity of the cell membrane in SH-SY5Y cells exposed to 5 µM Aβo and treated with Aβo + 1 µM curcumin and Aβo + 1 µM GT863. In the absence of 5 µM Aβo, the membrane-fluidity levels of control, 1 µM curcumin, and 1 µM GT863-treated cells were 1.21 ± 0.10 and 0.78 ± 0.075 ratio/µg protein (no significant difference, *n* = 10, Tukey). +: inclusion of 5 µM Aβo, 1 µM curcumin, 1 µM GT863; −: non-inclusion. The *p*-values in ANOVA were 0.0005. Each value expresses the mean + S.E.M. of at least 3 independent experiments. # *p* < 0.05, ## *p* < 0.01 for Aβo-exposed cells versus the other treated cells (*n* = 10, Tukey). (**C**–**N**) The SH-SY5Y cells observed using LipiORDER fluorescence. The respective excitation wavelengths were 402.9 nm, with green fluorescence wavelengths at 500–550 nm (**C**–**F**) and red fluorescence wavelengths at 570–620 nm (**G**–**J**). The green and red fluorescence images were analyzed ratiometrically (red/green) to visualize the lipid order (**K**–**N**). Lo; liquid-ordered phase, Ld; liquid-disordered phase. The scale bars represent 20 μm.

**Figure 7 ijms-24-03089-f007:**
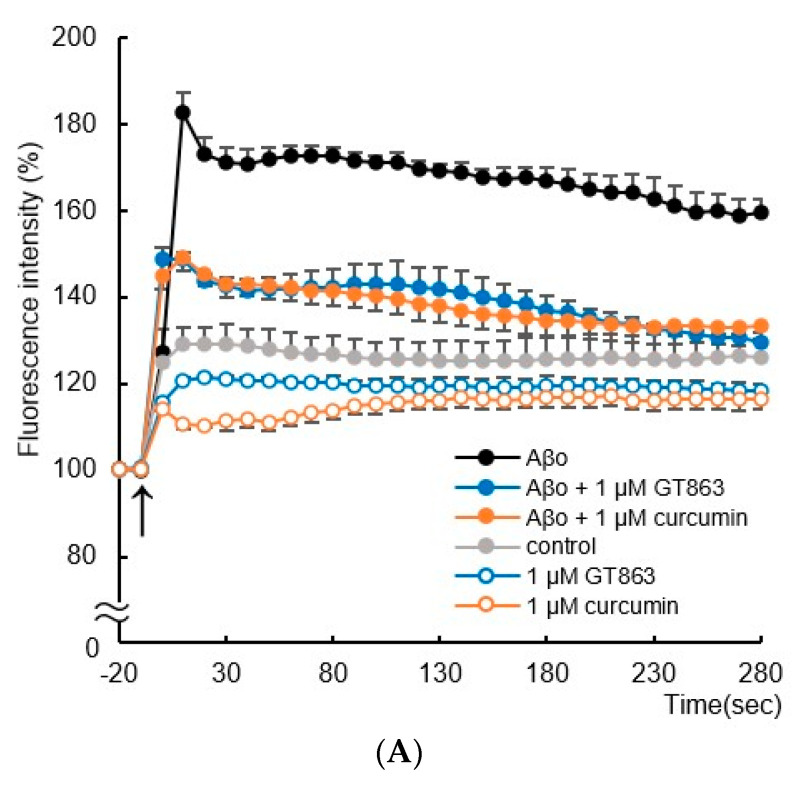
Effect of curcumin and GT863 on cell-membrane potential and electrical resistance of the cell membrane. (**A**) The membrane potential in Aβo-stimulated SH-SY5Y cells was evaluated using DiBAC_4_(3). The membrane potential in SH-SY5Y cells exposed to 5 µM Aβo and treated with Aβo + 1 µM curcumin and Aβo + 1 µM GT863 (*n* = 4, Tukey’s test). (**B**) Resting-membrane potential was measured in SH-SY5Y cells treated with 5 µM Aβo, curcumin, and GT863 by patch-clamp recording. ## *p* < 0.01 vs. Aβo group. (**C**) Electrical resistance of the membrane was measured in SH-SY5Y cells treated with 5 µM Aβo, curcumin, and GT863 by patch-clamp recording. # *p* < 0.05 vs. Aβo group. ## *p* < 0.01 vs. Aβo group. The *p*-values in ANOVA were 0.0001. Each value expresses the mean ± S.E.M. of at least 3 independent experiments.

**Figure 8 ijms-24-03089-f008:**
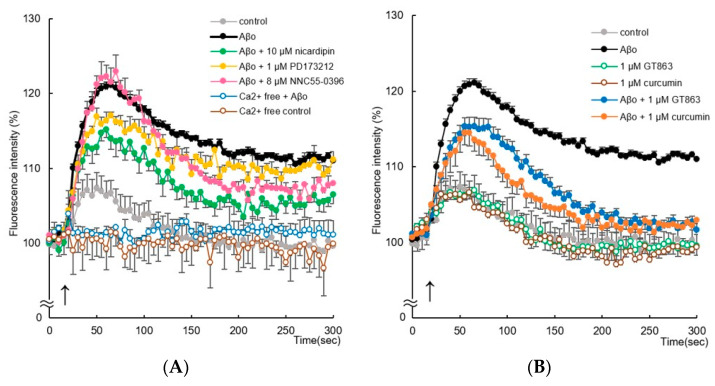
Effect of curcumin and GT863 on intracellular ionized-calcium concentration ([Ca^2+^]i) in SH-SY5Y cells. (**A**) Changes in [Ca^2+^]i were measured by the fluorescence intensity of SH-SY5Y cells exposed to 5 µM Aβo and calcium-channel antagonists (10 µM nicardipine, 1 µM PD173212, and 8 µM NNC55-0396) for up to 300 min. (**B**) Changes in [Ca^2+^]I were measured by the fluorescence intensity of SH-SY5Y cells exposed to 5 µM Aβo and treated with curcumin and GT863. The time of administration is indicated by an arrow. The fluorescence intensity was evaluated with the value at onset as 100%. The *p*-values in ANOVA were 0.0001. Each value expresses the mean ± S.E.M. of at least 3 independent experiments (*n* = 6).

**Figure 9 ijms-24-03089-f009:**
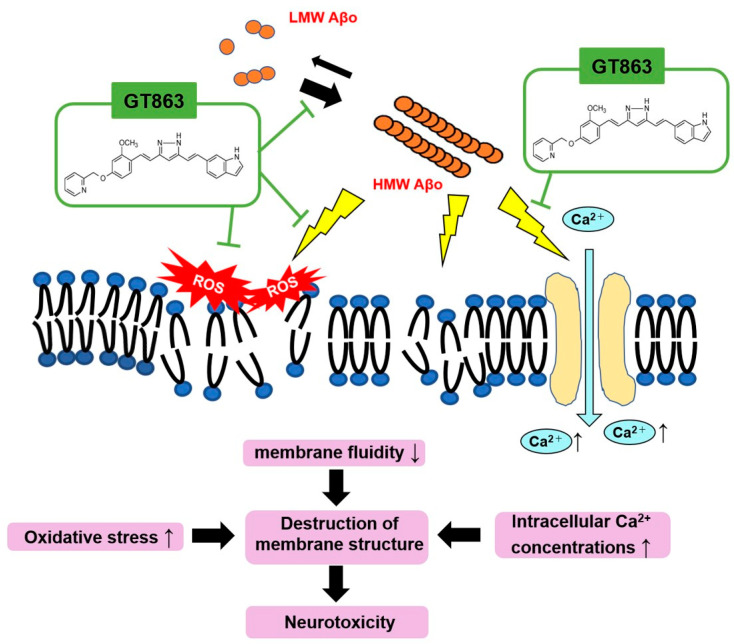
Schematic diagram of neuronal-membrane injury by Aβo and inhibition by GT863. Aβo injures neurons by increasing oxidative stress in the cell membrane, reducing cell-membrane fluidity and enhancing Ca^2+^ influx into the cell via VGCC. GT863 not only inhibits the promotion of Aβ polymerization but also prevents Aβo-induced damage to cell membranes.

## Data Availability

Not applicable.

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
