# Peer review of "The Curcumin Derivative GT863 Protects Cell Membranes in Cytotoxicity by Aβ Oligomers"

_ijms, 2023, doi:10.3390/ijms24043089_

Round 1
Reviewer 1 Report
This study entitled "The curcumin derivative GT863 protects cell membranes in cytotoxicity by Aβ oligomers" demonstrated the mechanism by which GT863, a derivative of curcumin with higher bioavailability, protects against the neurotoxicity of amyloid beta protein (Aβ) oligomers in human neuroblastoma cells. The study focused on how GT863 protect cell membrane damage caused by Aβ oligomers, including membrane fluidity, resistance, and changes in intracellular calcium.
The paper is well-written and well-researched, and I believe it contributes significantly to guide in AD research. However there are some suggestions to make this paper better:
1) Author monitored the aggregation kinetics every 15 min and showed Aβ1-40 for 360 min and Aβ1-42 for just 120 min, Is there any proper explanation for choosing different time interval? How was the graph of Aβ1-42 in 360 min? In comparative study the parameters should be same such as time interval and concentrations. Suggested to show the result with same time interval or mention the reason to choose different time point.
2) For result showing "Effects of curcumin and GT863 on viability and neurotoxicity in Aβo-exposed cells" it would be better to show the IC50 value of Aβo in Figure 3A and correct the Figure 3A graph alignment.
3) Figure 4A, showed Aβo alone is highly toxic to SH-SY5Y cells but in phase-contrast images the cell numbers seems different in Aβo+curcumin and Aβo+GT863 as compared with control and Aβo alone. The representative images are not matched with your graph and result explanations. Why the confluency of cells are different in Figure 4 B-E? If we observed the phase-contrast images only it seems that curcumin and GT863 treatment caused cell loss as compared to control and Aβo only. It would make more sense if cell number are less in Aβo group. Please choose the appropriate representative figure while showing in the paper.
4) Same issue with Figure 5, the cell confluency are very different seems Aβo+curcumin and Aβo+GT863 causing cell death as compared with control and Aβo. Also fluorescence intensity showed in graph 5A and 5F-I are not matching with each other. If we observe figure 5G the images showed 100% fluorescence as compared with other treatment group while graph 5A showed not much difference just around 20 unit. Suggested to show appropriate representative images that match your graph and result explanation.
Author Response
Manuscript ID: ijms- 2179394
Reply to Reviewers
Dear Reviewer #1,
Thank you very much for your mail of 17 January 2023 forwarding us reviewers’ comments on our manuscript titled “The curcumin derivative GT863 protects cell membranes in
cytotoxicity by Aβ oligomers” (Manuscript ID: ijms-2179394). Please accept our sincere gratitude to you and two respectable reviewers, who kindly advised us many points to be improved. We considered each point carefully and corrected them as advised.
The answer to each comment is shown in the following:
- Author monitored the aggregation kinetics every 15 min and showed Aβ1-40 for 360 min and Aβ1-42 for just 120 min, Is there any proper explanation for choosing different time interval? How was the graph of Aβ1-42 in 360 min? In comparative study the parameters should be same such as time interval and concentrations. Suggested to show the result with same time interval or mention the reason to choose different time point.
→Thank you very much for your comment. The ThT fluorescence intensity of Aß1-42 incubated under aggregation conditions showed a maximum peak of fluorescence after 100 min of aggregation but Aβ1-40 aggregation reached its maximum after 330 min. Aβ1-42 is known to aggregate faster than Aβ1-40. Furthermore, a decrease in fluorescence values is observed after the maximum peak of fluorescence. In order to observe the action of curcumin and GT863 on the respective Aβ aggregation in this experiment, the measurement time was set to the maximum peak of fluorescence.
- For result showing "Effects of curcumin and GT863 on viability and neurotoxicity in Aβo-exposed cells" it would be better to show the IC50 value of Aβo in Figure 3A and correct the Figure 3A graph alignment.
→Thank you very much for your comment. The IC50 value of Aβo was calculated and was 3.26µM. We also corrected the Fig. 3-A graph alignment.
Line 129: ” The IC50 value for cell viability of Aβo-exposed SH-SY5Y cells for 3 hr was 3.26 µM.”
- Figure 4A, showed Aβo alone is highly toxic to SH-SY5Y cells but in phase-contrast images the cell numbers seems different in Aβo+curcumin and Aβo+GT863 as compared with control and Aβo alone. The representative images are not matched with your graph and result explanations. Why the confluency of cells are different in Figure 4 B-E? If we observed the phase-contrast images only it seems that curcumin and GT863 treatment caused cell loss as compared to control and Aβo only. It would make more sense if cell number are less in Aβo group. Please choose the appropriate representative figure while showing in the paper.
→Thank you very much for your comment. As you indicated, we have changed to the different appropriate representative image in Fig. 4.
- Same issue with Figure 5, the cell confluency are very different seems Aβo+curcumin and Aβo+GT863 causing cell death as compared with control and Aβo. Also fluorescence intensity showed in graph 5A and 5F-I are not matching with each other. If we observe figure 5G the images showed 100% fluorescence as compared with other treatment group while graph 5A showed not much difference just around 20 unit. Suggested to show appropriate representative images that match your graph and result explanation.
→Thank you very much for your comment. As you indicated, we have changed the image to the appropriate representative image consistent with the results in Figure 5A.
Reviewer 2 Report
Major comments:
1. It is unclear what the authors refer to as Wieslab diagnostic services. The reference stated refers to ‘gbd dementia forecasting collaborators’. Please use the appropriate description.
2. Figure 1: Aβ aggregation kinetics does not show any error bars, data on ‘N’s or discuss statistical test for significant differences. This information needs to be incorporated.
3. Figure 7&8- provide error bars for Figure 7A, 8a,8b. Specify the”n” under figure legend and not under results section 2.4.5 or 2.5
4. Please provide values for Curcumin only and GT863 only controls for figure 8. Also provide quantification for figure 8.
5. In the discussion authors mention that GT863 may reduce probability of Aβo binding to the membrane. Please provide alternate explanations. Is there evidence of direct binding of GT863 to Aβ species and thus sequestration of Aβ or change in its conformation or disintegration of existing Aβ oligomer/protofibrils etc?
6. Authors state that, nicardipine pretreatment did not inhibit Ca2+ intracellular influx by 100%, however, the traces is 8A for Control and Aβ+ Nicardipine appear very close together suggesting almost complete inhibition. In this scenario quantification and bar graphs would be necessary to support the claims. Also, what is the rationale for choosing the dosage of various inhibitors? Moreover, to test whether GT863 inhibits Ca++ influx via VGCC, GT863 dose response with and without nicardipine is necessary. This experiment will address the mode/mechanisms of GT863-mediated inhibition of Ca++ entry.
7. Also, GT863-mediated inhibition of Ca++ influx should be accompanied by a positive control of ionophores such as AT23187 to ascertain the presence of enough free Ca++ in the buffer, functionality and saturation of the dye under different treatment conditions.
8. The study gives interesting readouts about the protective effect of Curcumin and its derivatives on SHSY5Y cells. These results should be replicated on other neuronal cell lines and primary neuronal cultures. At least the authors should address this as limitations in the discussion section. Moreover, the authors should allude to the role of non-neuronal cell types mainly microglia and astrocytes in the CNS and briefly discuss the related literature or the lack thereof on the effects of Curcumin/GT863 on non-neuronal subtypes in the context of AD.
Minor Comments:
99. Figure 3, legend says Posthoc test as Turkey instead of Tukey.
110. Images in figure 4 and 5 appear blurry and a bit smaller. An image similar to figure 6 would be better.
111. Sentence is very complicated and needs to be re-written- The cell membrane of SH-SY5Y cells exposed to Aβo (Fig. 6-D, H, L) showed a higher 233 proportion of Lo phase with higher lipid membrane density from Ld phase with lower 234 lipid membrane density than the control.
112. Grammatical errors in a few places need to be corrected. Line 337, 532
113. A reference is missing for the anti-Aβo action of aducanumab in the following sentence- targeting Aβ or its aggregates have been pursued, and aducanumab an anti-Aβo antibody. Also, Aducanumab acts against Ab aggregates including oligomers and insoluble fibrils. The sentence needs to be restructured to reflect that.
114. It is unclear what the authors refer to as ‘another structure’ in the following sentence-
However, in GT863, the β-diketone structure is 380 replaced by a pyrazole group, and the phenol group is replaced by another structure. Line 381
115. In line 389 please write Aβ species as opposed to Aβs.
116. Arrows in Figure 9 should be made bigger.
117. I think authors refer to Sprague Dawley rats in line 450. That needs to be spelled out.
Author Response
Manuscript ID: ijms- 2179394
Reply to Reviewers
Dear Reviewer #2,
Thank you very much for your mail of 17 January 2023 forwarding us reviewers’ comments on our manuscript titled “The curcumin derivative GT863 protects cell membranes in
cytotoxicity by Aβ oligomers” (Manuscript ID: ijms-2179394). Please accept our sincere gratitude to you and two respectable reviewers, who kindly advised us many points to be improved. We considered each point carefully and corrected them as advised.
The answer to each comment is shown in the following:
Major concerns
- It is unclear what the authors refer to as Wieslab diagnostic services. The reference stated refers to ‘gbd dementia forecasting collaborators’. Please use the appropriate description.
→Thank you very much for your comment. As you indicated, we have revised the text and changed the references as follows
Lines 41-43: “Recent estimates predict that 50.4 to 65.1 million people worldwide were affected by dementia in 2019, with the prevalence increasing to 130.8 to 175.6 million by 2050 [1].”
Lines 690-692 (Reference 1): ”G.B.D. Dementia Forecasting Collaborators. Estimation of the global prevalence of dementia in 2019 and forecasted prevalence in 2050: an analysis for the Global Burden of Disease Study 2019. Lancet Public Health 2022. 2022, 7, e105-25. https://doi.org/10.1016/S2468-2667(21)00249-8.”
- Figure 1: Aβ aggregation kinetics does not show any error bars, data on ‘N’s or discuss statistical test for significant differences. This information needs to be incorporated.
→Thank you very much for your comment. We agree with your comment and have corrected the information by adding the missing information. We added the error bars, the number of “n”, and statistical test for significant differences in Figure 2.
- Figure 7&8- provide error bars for Figure 7A, 8a,8b. Specify the”n” under figure legend and not under results section 2.4.5 or 2.5
→Thank you very much for your comment. As well as your point above, we have corrected the information by adding the missing information. We have added in the error bars for Figure 7(A) & Figure 8(A) (B) and the number of "n" under their figure legends.
- Please provide values for Curcumin only and GT863 only controls for figure 8. Also provide quantification for figure 8.
→Thank you very much for your comment. We append in Fig. 8(b) the [Ca2+]i changes when treated with 1 µM curcumin only and 1 µM GT863 only. The statistical results of Fig. 8 have been added to the results section.
- In the authors mention that GT863 may reduce probability of Aβo binding to the membrane. Please provide alternate explanations. Is there evidence of direct binding of GT863 to Aβ species and thus sequestration of Aβ or change in its conformation or disintegration of existing Aβ oligomer/protofibrils etc?
→Thank you very much for a very meaningful advice. As you point out, there is no evidence that GT863 binds directly to Aβ or that it disintegrates oligomers. However, GT863 inhibited Aβ aggregation, suggesting that GT863 may bind to Aβ and restore membrane fluidity by inhibiting the action of Aβo on the membrane. In our next experiments, we would like to confirm that GT863 binds directly to Aβ, resulting in Aβ containment, conformational changes and disruption of existing Aβ oligomers.
Please let us express our great gratitude to you for this encouraging and pertinent advice.
We rewrote the results description in the Results section, lines 421-423.
“Considering that GT863 also strongly inhibited Aβ aggregation (Fig. 2), it is possible that GT863 binds to Aβo, inhibiting the direct effect of Aβo on the membrane and improving the loss of fluidity.”
- Authors state that, nicardipine pretreatment did not inhibit Ca2+ intracellular influx by 100%, however, the traces is 8A for Control and Aβ+ Nicardipine appear very close together suggesting almost complete inhibition. In this scenario quantification and bar graphs would be necessary to support the claims. Also, what is the rationale for choosing the dosage of various inhibitors? Moreover, to test whether GT863 inhibits Ca++ influx via VGCC, GT863 dose response with and without nicardipine is necessary. This experiment will address the mode/mechanisms of GT863-mediated inhibition of Ca++ entry.
→Thank you very much for your comment. For nicardipine, we corrected the data to 10 µM instead of 20 µM, as other literature has experimented with 10 µM. We sincerely apologize for the flaw in the data. The rationale for selecting the inhibitor doses was based on the following references, respectively.
Nicardipine: Yasumoto T. et al. FASEB J. 2019 (PMID: 31084283). In this literature, experiments were performed in SH-SY5Y cells at 10 µM and a calcium channel blocking effect was obtained.
NNC55-0396: Li M. et al. Cardiovasc Drug Rev. 2005 (PMID: 16007233). Although the cells are HEK293á1G cells, not SH-SY5Y cells, it is stated that "NNC 55-0396 blocked more than 50% of the T-type Ca2+ current at 8 µM".
PD173212: Coppi E. et al. Pain. 2019 (PMID: 31008816). In this literature, PD173212 is tested at 0.5 µM and 1 µM. Furthermore, the product description states that "PD173212 is a selective N-type voltage sensitive calcium channel (VSCC) blocker, with an IC50 of 36 nM in IMR-32 assays. (PMID: 10465535)” The channel blocking effect was not obtained at the 1 µM concentration we tested.
As you pointed out, we do not know whether GT863 acts on the calcium channel in this experiment. However, since GT863 inhibits Aβ aggregation, we rethink that GT863 binds to Aβ and inhibits calcium influx through VGCCs by Aβ.
Lines 438-441: “As shown in Fig. 2, GT863 inhibited Aβ aggregation, suggesting that GT863 bound to Aβo and inhibited the effects of Aβo on VGCCs. Therefore, we suggest that GT863 reduced cytotoxicity by suppressing the increase in intracellular Ca2+.”
- Also, GT863-mediated inhibition of Ca++ influx should be accompanied by a positive control of ionophores such as AT23187 to ascertain the presence of enough free Ca++ in the buffer, functionality and saturation of the dye under different treatment conditions.
→Thank you very much for your comment. We agree with your comment and wrote the experimental results using Ca2+-free buffer to Fig. 8 (A) and the Results section. In addition, the Materials and methods section has been updated.
- The study gives interesting readouts about the protective effect of Curcumin and its derivatives on SHSY5Y cells. These results should be replicated on other neuronal cell lines and primary neuronal cultures. At least the authors should address this as limitations in the discussion section. Moreover, the authors should allude to the role of non-neuronal cell types mainly microglia and astrocytes in the CNS and briefly discuss the related literature or the lack thereof on the effects of Curcumin/GT863 on non-neuronal subtypes in the context of AD.
→Thank you very much for your comment. As you pointed out, we have not been able to evaluate the effect in other cells, such as non-neuronal cell types, etc. We have added research references using other cells to the Introduction section and Discussion section.
Kato H. et al. Amyotroph Lateral Scler Frontotemporal Degener. 2022 (PMID: 34894926).
Mohammadi A. et al. Neuroimmunomodulation. 2022 (PMID: 34496365).
Line 73-74: “Furthermore, GT863 was reported to be neuroprotective against hydrogen perox-ide-induced cytotoxicity in PC12 cells [15].”
Lines 453-455: “Curcumin has been shown to be protective not only for neurons but also for astrocytes, which play an important role in brain homeostasis [42].”
Lines 477-479: “However, given the limitations of this study, further studies using different cell types, such as non-neuronal cell lines, are needed to confirm the efficacy of GT863.”
Minor concerns
- Figure 3, legend says Posthoc test as Turkey instead of Tukey.
→Thank you very much for your comment. We apologize for the simple spelling error. We have corrected it.
- Images in figure 4 and 5 appear blurry and a bit smaller. An image similar to figure 6 would be better.
→Thank you very much for your comment. We are very sorry that the image is difficult to see due to my lack of skill. Since there is no more visible image in this case, we will try to take pictures with care in the future.
- Sentence is very complicated and needs to be re-written- The cell membrane of SH-SY5Y cells exposed to Aβo (Fig. 6-D, H, L) showed a higher 233 proportion of Lo phase with higher lipid membrane density from Ld phase with lower 234 lipid membrane density than the control.
→Thank you very much for your comment. We apologize for making this text difficult to read. We rewrote the results description in the Results section, lines 239-243.
“The cell membrane of SH-SY5Y cells exposed to Aβo showed a decreased Ld phase as shown in Fig. 8-H and a lower ratiometric fluorescent value (red/green) as shown in Fig. 8-L than the control. Exposure to Aβo may have increased the lipid membrane density of the cell membrane, resulting in a stiffer cell membrane. This suggests that the density of fat in the cell membrane has increased, resulting in reduced fluidity.”
- Grammatical errors in a few places need to be corrected. Line 337, 532
→Thank you very much for your comment. Thank you for pointing this out. We have corrected the grammatical errors.
Lines 347-349: “In the present study, comparable to curcumin, GT863 showed dose-dependent inhibition of Aβ1-40 and Aβ1-42 aggregation at low concentrations (Fig. 2).”
Lines 546-547: “After incubation, the MTT assay was performed and measured at 570 nm using a microplate reader Spectra Max i3 (Molecular Devices).”
- A reference is missing for the anti-Aβo action of aducanumab in the following sentence- targeting Aβ or its aggregates have been pursued, and aducanumab an anti-Aβo antibody. Also, Aducanumab acts against Ab aggregates including oligomers and insoluble fibrils. The sentence needs to be restructured to reflect that.
→Thank you very much for your comment. Since lecanemab was recently approved by the FDA, we have changed the drug from aducanumab to lecanemab and revised the text as you have indicated.
Lines 52-57: “Therapeutic strategies targeting Aβ or its aggregates have been pursued, and lecanemab, an anti-Aβo antibody, was rapidly approved by the United States Food and Drug Ad-ministration (US FDA) in 2023. Recently, high molecular weight (HMW) Aβo mainly composed of protofibrils, which is a toxic Aβo, have attracted attention [5,6]. Lecanemab binds Aβ aggregates including oligomers and insoluble fibrils [7].”
- It is unclear what the authors refer to as ‘another structure’ in the following sentence-
However, in GT863, the β-diketone structure is 380 replaced by a pyrazole group, and the phenol group is replaced by another structure. Line 381
→Thank you very much for your comment. We have revised the text by stating the name of the unclear structures.
Lines 389-391: “However, in GT863, the β-diketone structure of curcumin is replaced by a pyrazole group, and the two phenol groups are replaced by an indol ring and a pyridylmethoxy group.”
- In line 389 please write Aβ species as opposed to Aβs.
→Thank you very much for your comment. Certainly, "Aβ species" is a more descriptive term. Thank you very much for your precise guidance.
Lines 334-336: “HMW Aβo are widely suggested to be the most toxic of the Aβ species and have been proposed as one of etiologies of AD [4].”
- Arrows in Figure 9 should be made bigger.
→Thank you very much for your comment. We apologize for the lack of visibility. We have enlarged the arrows as you indicated.
- I think authors refer to Sprague Dawley rats in line 450. That needs to be spelled out.
→Thank you very much for your comment. We are very sorry for using abbreviations without explanation. We have corrected it as you indicated.
Reviewer 3 Report
1. How the curcumin analog GT863 was prepared? What about their purity and characterization?
2. Explain some the important biological potentials of curcumin analogues (reported previously) in the introduction section.
3. Curcumin was used as a reference drug by the authors. why? I believe that some standard clinical drug candidate should be used as a reference.
Author Response
Manuscript ID: ijms- 2179394
Reply to Reviewers
Dear Reviewer #3,
Thank you very much for your mail of 17 January 2023 forwarding us reviewers’ comments on our manuscript titled “The curcumin derivative GT863 protects cell membranes in
cytotoxicity by Aβ oligomers” (Manuscript ID: ijms-2179394). Please accept our sincere gratitude to you and two respectable reviewers, who kindly advised us many points to be improved. We considered each point carefully and corrected them as advised.
The answer to each comment is shown in the following:
- How the curcumin analog GT863 was prepared? What about their purity and characterization?
→Thank you very much for this comment. The GT863 adjustment is described in the following paper.
Okuda M. et al. Bioorg Med Chem Lett. 2016 (PMID: 27624076).
GT863 replaced the two aromatic rings of curcumin with another structure, elevating its Aβ and tau aggregation inhibitory and metabolic protective effects. Furthermore, the central diketone group was replaced with a pyrazole group, increasing its stability in aqueous solution. GT863 is characterized by its higher brain translocation compared to curcumin.
GT863 describes what structural formula was changed from curcumin, lines 389-391.
“However, in GT863, the β-diketone structure of curcumin is replaced by a pyrazole group, and the two phenol groups are replaced by an indol ring and a pyridylmethoxy group.”
The purity is 99.1% and has been added to the Materials and methods section.
- Explain some the important biological potentials of curcumin analogues (reported previously) in the introduction section.
→Thank you very much for this comment. We have revised the introduction by adding text about curcumin analogues and derivatives. We also added references to the literature introducing curcumin analogues and derivatives to the References section.
Chainoglou E. et al. Int J Mol Sci. 2020 (PMID: 32183162).
Line 67-70: “These results have led to the synthesis of curcumin analogues and derivatives that are expected to be very useful drugs for the treatment of AD [11]. GT863 (Fig. 1) was synthesized as a derivative of curcumin; this derivative has higher bioavailability and brain translocation than curcumin [12].”
- Curcumin was used as a reference drug by the authors. why? I believe that some standard clinical drug candidate should be used as a reference.
→Thank you very much for this comment. The reason curcumin was used as the reference drug is because GT863 is a derivative of curcumin. Furthermore, we wanted to see if it was non-inferior to curcumin.
Regarding drugs currently used in clinical practice, there is literature showing that donepezil, an AChE inhibitor, is neuroprotective against toxicity due to depolarization, although it was not used in this experiment. It also appears to inhibit Ca2+ and Na+ influx into the cells, and it would be very interesting to compare this effect with that of clinical agents, which we would like to do in future experiments.
Akasofu S. et al. Eur J Pharmacol. 2008 (PMID: 18508044).
Round 2
Reviewer 2 Report
I think the authors have incorporated the suggestions satisfactorily.
Reviewer 3 Report
Manuscript is ok now